# Handheld PET Probe for Pediatric Cancer Surgery

**DOI:** 10.3390/cancers14092221

**Published:** 2022-04-29

**Authors:** Hannah N. Rinehardt, Sadie Longo, Ryan Gilbert, Jennifer N. Shoaf, Wilson B. Edwards, Gary Kohanbash, Marcus M. Malek

**Affiliations:** 1Department of General Surgery, University of Pittsburgh Medical Center, Pittsburgh, PA 15213, USA; 2Department of Surgery, University of Pittsburgh School of Medicine, Pittsburgh, PA 15213, USA; longose@upmc.edu (S.L.); gilbert.ryan@medstudent.pitt.edu (R.G.); 3Division of Pediatric Radiology, UPMC Children’s Hospital of Pittsburgh, Pittsburgh, PA 15224, USA; jennifer.conver@chp.edu; 4Department of Biochemistry, University of Missouri, Columbia, MO 65201, USA; wbe2@pitt.edu; 5Department of Neurological Surgery, University of Pittsburgh, Pittsburgh, PA 15201, USA; gary.kohanbash2@chp.edu; 6Division of Pediatric General and Thoracic Surgery, UPMC Children’s Hospital of Pittsburgh, Pittsburgh, PA 15224, USA

**Keywords:** FDG, PET, pediatric cancer, PTLD, Hodgkin’s lymphoma, radio-guided surgery, gamma probe, occult tumor, neuroblastoma, solid tumor

## Abstract

**Simple Summary:**

Positron emission tomography (PET)/computed tomography (CT) scans are widely used as a form of full body imaging and allow for the early detection of small, asymptomatic tumors that may represent cancer metastasis or recurrence. Tissue diagnosis is critical in determining the choice of ongoing targeted therapy for pediatric patients with solid tumors. These small tumors may be difficult to localize in the operating room, especially in a re-operative or radiated area of the body. An adjunct such as a PET probe, used to guide intra-operative dissection, is the ideal tool to assist in cases where an occult tumor requires an excisional biopsy.

**Abstract:**

18F-fluorodeoxyglucose (FDG) is a glucose analog that acts as a marker for glucose uptake and metabolism. FDG PET scans are used in monitoring pediatric cancers. The handheld PET probe localization of FDG-avid lesions is an emerging modality for radio-guided surgery (RGS). We sought to assess the utility of PET probe in localizing occult FDG-avid tumors in pediatric patients. PET probe functionality was evaluated by using a PET/CT scan calibration phantom. The PET probe was able to detect FDG photon emission from simulated tumors with an expected decay of the radioisotope over time. Specificity for simulated tumor detection was lower in a model that included background FDG. In a clinical model, eight pediatric patients with FDG-avid primary, recurrent or metastatic cancer underwent a tumor excision, utilizing IV FDG and PET probe survey. Adequate tissue for diagnosis was present in 16 of 17 resected specimens, and pathology was positive for malignancy in 12 of the 17 FDG-avid lesions. PET probe gamma counts per second were higher in tumors compared with adjacent benign tissue in all operations. The median ex vivo tumor-to-background ratio (TBR) was 4.0 (range 0.9–12). The PET probe confirmed the excision of occult FDG-avid tumors in eight pediatric patients.

## 1. Introduction

Diagnostic whole-body 18F-fluorodeoxyglucose (FDG) positron emission tomography computed tomography (PET/CT) imaging is used to identify and assess metabolically active tissue. Rapidly dividing cancer cells have a high number of glucose transporters and high rates of glycolysis, increasing the uptake of FDG [1]. FDG is a radioisotope with a half-life of 110 min, and its decay involves the release of high-energy 511 keV directional photons [2]. PET/CT is highly sensitive to detecting a variety of cancers, including childhood cancers such as Hodgkin’s lymphoma, neuroblastoma, and post-transplant lymphoproliferative disorder (PTLD) [3,4]. PET/CT may be used for initial staging, in monitoring for recurrence or metastasis, or to identify occult residual disease after tumor resection [3,4]. When performing an excisional biopsy of recurrent disease or metastasis, small tumors can be difficult to localize intra-operatively, particularly when in a post-operative or radiated field. Novel techniques to assist with tumor identification in the operating room have the potential to identify regions of occult disease and facilitate a safer and more complete excision of residual or recurrent disease.

Radio-guided surgery (RGS) utilizing technetium-99 m and a handheld gamma probe was first described in 1981 and has become the standard of care for sentinel lymph node biopsy [5]. The use of FDG and a handheld gamma probe specifically designed for high-energy gamma ray detection (PET probe) for RGS was first described in 1999, but its intraoperative use has not become routine [5]. The reported evidence in adult cancers includes heterogeneous case series [6,7,8,9,10]. PET probe performance has shown to be dependent on a variety of factors including anatomic location, time from injection to probe survey and physical distance between the probe and tumor. The PET probe’s major limitation is poor specificity as its use is confounded by high gamma detection from adjacent metabolically active organs. To assess the specificity of the probe to malignant tumors of interest, a tumor to background ratio (TBR) in situ and ex vivo is often calculated. TBR is a simple calculated ratio of gamma counts per second (cps) of a tumor divided by gamma cps of non-lesional tissue (background tissue). If the tumor has a higher gamma cps than the background, the TBR will be >1. A higher TBR is associated with a higher specificity of the detection system.

The absolute threshold for an adequate TBR has yet to be established in operative use of the PET probe. In 2000, Yasuda et al. described that a minimum TBR of 5 is required to distinguish an FDG-avid tumor from a background using the handheld gamma probe in vitro [11]. In 2001, Zervos et al. demonstrated the successful use of a PET probe in the detection of recurrent colorectal cancer. Ten patients found to have FDG-avid lesions in a pre-op PET/CT then underwent resection with PET probe use, and post-op pathology confirmed the resection of malignant lesions. In this study, the mean in situ TBR was 1.5 [6]. In 2007, Hall et al. described intra-operative use of a PET probe to confirm the complete resection of metastatic breast cancer in two patients [7]. In 2008, Cohn et al. demonstrated the use of a PET probe for the detection of recurrent epithelial ovarian cancer in three patients [8]. Povoski described use of the probe in the resection of three metastatic melanoma lesions in one patient in 2008 and in thirteen patients with lymphoma in 2015 [9,10]. TBR values were not reported in some studies.

Other options are available for intraoperative tumor localization. The literature demonstrates the successful localization of occult pulmonary lesions in pediatric patients using a variety of methods including CT-guided coil or wire placement. CT-guided microcoil placement is useful to guide surgeons to find small pulmonary nodules [12,13]. CT-guided wire localization is another useful adjunct for localizing pulmonary nodules but also risks pneumothorax, increased time under anesthesia and inadvertent wire dislodgement [14]. The use of methylene blue dye has also been reported as an important adjunct for thoracoscopic tumor localization [13,14]. Magnetic tracer placement by ultrasound-guidance and a magnetic probe for the detection of nonpalpable breast lesions is another method of occult tumor localization that has been described with some success [15]. Ultrasound-guided magnetic tracer localization avoids the risk of radiation exposure to patients and perioperative staff associated with the use of radiotracers and CT-guided methods. The methods mentioned above, require a lesion that is clear in imaging, with the placement of a localizing agent adjacent to it that will allow for the intraoperative detection of the lesion. PET-avid lesions can be small, and at times, may have FDG avidity as their only distinguishing characteristic on imaging. Additionally, some lesions are not accessible via percutaneous techniques. Thus, methods such as wire, microcoil and magnetic tracer localization would not be possible for many of the tumors in our study, particularly those in the abdomen and mediastinum.

The use of RGS in pediatric patients with a radiotracer was first described in 1997 by Heij et al. utilizing I^123^-MIBG-directed surgery to aid the resection of neuroblastoma in five patients [16]. There have been a few subsequent reports demonstrating the use of intraoperative gamma probe detection of neuroblastoma, but the largest such series, reported that I^123^-MIBG was not helpful in 35% of cases, with a specificity for malignant tissue of only 55% [17,18,19,20]. Given the limited specificity of the gamma probe and variable results from prior studies, SLNB remains the only regular use of RGS in pediatric surgery. Despite the widespread usage of FDG for preoperative imaging, there are no prior studies examining FDG for RGS in pediatric cancer. We seek to define the baseline performance of the PET probe in a PET scan calibration model and in a clinical model of RGS. Ultimately, we aim to evaluate PET probe RGS as a modality to facilitate the early identification and treatment of recurrent or metastatic childhood cancers.

## 2. Materials and Methods

### 2.1. PET/CT Calibration Phantom Design

In a preclinical setting, a PET/CT calibration phantom model was used. This model was developed by the Clinical Trials Network as a method to validate PET/CT scanners for use in oncology clinical trials [21]. The phantom contains six spheres with diameters ranging from 0.7 to 2.0 cm that are filled with a solution of concentrated FDG of 24.0 kBq/mL. The spheres are situated within a fluid-filled thoracic cavity containing dilute FDG of 6.0 kBq/mL. The ratio of FDG in the solution relative to the spheres should produce a standardized uptake value (SUV) of 4 in the spheres on PET/CT 60 min after fill time. This protocol was recapitulated, and PET scan was followed by PET probe survey to simulate the planned operative procedures (Figure 1). For comparison, in a separate experiment, the spheres were filled with the same concentration of FDG, while the background cavity was filled with saline-only spheres. Gamma counts per second (cps) of each lesion were measured, and background readings from the fluid were obtained to calculate a TBR.

### 2.2. Inclusion Criteria and Participants

A clinical prospective analysis included children aged 21 years and under with suspected recurrent, primary, or metastatic PET-avid lesions who underwent tumor excision or biopsy utilizing intravenous (IV) 18F-FDG and handheld PET probe at the Children’s Hospital of Pittsburgh from 1 January 2018 to 1 March 2021. This study was approved by the Institutional Review Board (IRB) at the Children’s Hospital of Pittsburgh. The study included 9 patients who met the inclusion criteria. All participating patients underwent a pre-operative PET/CT within one month of the operation, which identified one or more occult PET-avid lesions. PET avidity was defined as SUV > 4. Tumors were defined as occult by the operating surgeon if they were expected to be difficult to find based on size and/or location.

### 2.3. Prospective Study Design

Based on prior studies, all patients received a same-day, single-dose preoperative IV injection of 18F-FDG 0.2 mCi/kg via a peripheral IV line one hour prior to the operation and approximately two hours prior to intra-operative probe use for tumor localization [5,20]. Per standard PET scan protocol, patients rested in a quiet, dark room in the interval between injection and transfer to the operating suite. All patients fasted for a minimum of 6 h prior to receiving the 18F-FDG injection. Care was taken to ensure that IV fluids administered perioperatively and intraoperatively were non-dextrose containing. The Neoprobe High Energy F-18 Probe (Mammotome, Cincinnati, OH, USA) was utilized for intra-operative tumor detection (Figure 2). This probe detects high-energy photons emitted during FDG decay and involves sophisticated internal shielding to enhance directionality. An external survey was performed to determine external values for several organs (body regions), including the kidneys (flanks), spleen (posterior left upper quadrant), liver (right upper quadrant under costal margin), bladder (suprapubic), and brain (top of scalp). External values of the distal extremity and room background were also measured for comparison. External surveys were performed during induction of anesthesia to prevent prolonging the operative procedure. During open cases, intra-operative probe guidance was attempted. A disposable sterile probe cover was utilized. The lesion of interest was excised, and the handheld probe was used to measure gamma counts in the lesion ex vivo. Adjacent non-lesional tissue (of similar size) was also evaluated to calculate a TBR. Final pathology reports of the 17 excised lesions were reviewed.

### 2.4. Outcomes and Statistical Analysis

The primary outcome of interest to evaluate PET probe utility in tumor identification is TBR. TBR was calculated in preclinical and clinical models. Data including demographics, gamma cps measurements, and TBR were reported with median values and ranges or means and standard deviations. TBR values between malignant and benign groups are compared with a nonparametric Mann–Whitney U test given the small sample size in this study (*n* = 17). A *p*-value < 0.05 is used as a threshold for statistical significance.

## 3. Results

### 3.1. Phantom Model Performance

In the thoracic phantom model (Figure 1), the handheld PET probe localized all six FDG-containing spheres when background radioactivity was absent (saline alone), confirming the ability of the probe to detect the high-energy gamma photons produced by FDG decay. The cps associated with the simulated tumors was measured at several time points resulting in a decreasing pattern as expected for the decay of FDG (Figure 3). Next, the phantom’s background compartment was filled with dilute FDG, and an SUV of 4 in the lesions was confirmed on PET/CT prior to PET probe survey. Measurements of gamma counts for each lesion and the background were recorded at 110 min from experiment start time and used to calculate TBR (Table 1). Median TBR for the spheres was 1.07 (range 1.06–1.11) at 110 min in a dilute FDG background.

### 3.2. Patient Demographics

Nine patients were included in the study. For the analysis, one patient who received IV I^123^-MIBG prior to surgery for localization of neuroblastoma was excluded. Eight patients met criteria for the data analysis. Within the patient population (*n* = 8), a total of 17 FDG-avid tumors were identified by pre-operative PET/CT scans. The median age was 16 years old (range 5–21 years old). Of the eight patients, six were male and two were female. The diagnoses included three patients with suspected PTLD, three with Hodgkin’s Lymphoma, one with Burkitt’s lymphoma, and one with neuroblastoma. Patient demographics are illustrated in Table 2. Pre-op PET/CT was performed within one month of the operation with a median of 10 days (range 2–29 days). One pre-op PET/CT median tumor SUV was 7 (range 3.8–16.7).

### 3.3. External Survey

A handheld gamma probe was used to measure external readings from surrounding organs for the assessment of background FDG uptake (Table 3). An external survey was performed in eight of the nine operations in the study at a median of 65 min after injection of IV FDG (range 20–175 min). Areas overlying the liver, spleen, kidneys, brain, and the distal extremity were evaluated with a mean >300 from the liver, spleen, kidney, and brain external surveys. Distal extremity and room background had a low median cps of 50 and 2, respectively.

### 3.4. Intra-Operative Probe Performance

Probe survey of excised lesions was performed at a median of 101 min after IV FDG injection (range 65–210 min). PET probe data were collected from the prospective evaluation of nine pediatric cancer operations and in eight patients with excision of seventeen FDG-avid lesions, including ex vivo cps with calculated TBR. Eight patients underwent surgical exploration, including five minimally invasive surgeries and four open operations for the removal of PET-positive lesions. One patient underwent two separate FDG-guided procedures to remove PET-positive lesions for PTLD. Seventeen lesions in total were excised from the nine operations included in this study (Table 4). Six lesions were excised from the abdomen, two from the mediastinum, five from the retroperitoneum (RP), four from the neck, and one from the lung. In open cases, the PET probe provided a poor lesion specificity and could not be used for intraoperative guidance in the identification of lesions in vivo. The probe is too large for use in laparoscopic and thoracoscopic cases; therefore, it was only used for ex vivo analysis. Lesion size ranges from 0.1 cm to 2.5 cm, with a median size of 2.0 cm. The final pathology was positive for the suspected malignancy in 12 of the 17 excised lesions. The median cps for the excised lesions was 32, and the mean cps for the background tissue was 11. The average ex vivo TBR for the lesions was 4.0 (range 0.9–12.0). When comparing the malignant and benign lesions based on final pathology, the average ex vivo TBR values were 4.7 and 2.0, respectively with a *p*-value of 0.0181 between the two groups.

## 4. Discussion

Gamma probes capable of detecting high-energy gamma emissions (PET probes) have been available for over 20 years [5]. Despite the high prevalence and clinical relevance of PET scanning, PET probes have not found a routine place in cancer surgery. This study looked to define the utility and limitations of PET probes and describe their first use in pediatric patients. This study also provides the first patient-level data on the external measurements of FDG gamma emissions from various regions of the body (Table 3).

The preclinical thoracic phantom model confirmed the ability of the PET probe to detect high-energy gamma emission from FDG decay. This was an important step in validating the probe for gamma detection with the well-established half-life of the radioisotope prior to trialing clinical use of the probe. Of note, in Figure 3, sphere 1 emitted a higher gamma cps compared with other simulated lesions. Sphere 1 was the second-largest sphere (1.5 cm in size) and was closest to the phantom’s surface, which we suspect was the reason for the high recorded values. When background levels of FDG were added to the phantom, the median TBR was only 1.07, indicating an inability of the PET probe to localize simulated lesions. It is important to note that prior studies described that successful localization requires a TBR of at least 1.5 [6,11,22]. The preclinical phantom model was critical for probe validation as our clinical study did not have a control arm.

The patients in our series had small tumors with a minimum SUV of 4 and a median SUV of 7.0. Background FDG avidity was confirmed by an external survey and intra-operative probe use. Metabolically active organs (kidneys, spleen, liver, and brain) by an external survey demonstrated mean gamma cps > 300. This correlated well with typical PET scan findings, which note a high uptake in these metabolically active organs. This is the first description, however, of gamma activity recordings via an external survey with a handheld PET probe. Not surprisingly, the highest cps are noted at the bladder, which is a result of FDG excretion via the kidneys into the urine. The second highest cps are recorded at the brain given the high glucose requirement for neural tissue metabolism. These findings indicate a potential reason for the PET probe’s lack of specificity, as FDG uptake is seen throughout the body, with a very high accumulation in metabolically active organs. Values from these benign organs were at least 300% greater than the measurements from the small, malignant tumors. External survey data from our experiment certainly indicated that eliminating background signals would be important to allow for the routine intraoperative use of the PET probe for tumor localization. This knowledge is critical for guiding future studies that may aim to improve probe specificity and highlights the importance of tumor-specific markers as a potential future research direction.

Fluorescent-guided surgery is gaining enthusiasm as a method for tumor-specific surgical guidance. The most commonly used agent, indocyanine green (ICG), is actually a non-specific, water-soluble, near-infrared (NIR) dye that can accumulate in some lesions such as lung and liver tumors [23]. Targeted NIR agents, have the advantage of allowing for a more specific detection, with less background signal. The use of an ICG-like fluorescent molecule conjugated to an anti-CEA antibody has been described in a preclinical model [24]. An OTL38 (folate analog conjugated to NIR dye) has been successfully used to target tumors that express folate receptors, such as in ovarian and lung cancers [25,26]. An anti-epidermal growth factor receptor (EGFR) antibody panitumumab conjugated to a NIR dye is under investigation and has been used in resection of head and neck cancers that express EGFR [27]. The tumors in the current study, however, were identified solely on the basis of their FDG avidity from a PET scan. Without a clear targeted agent to utilize, IV FDG was used to determine if FDG could guide surgical resection via PET probe detection.

Consistent with the preclinical model, the in vivo probe performance was impacted by a lack of specificity and was unable to provide surgical guidance to tumors. The PET probe was able to confirm FDG avidity ex vivo after suspected lesions were excised. The recorded median ex vivo cps of the small FDG-avid tumors (17 lesions) was 32, compared with the median ex vivo cps of the adjacent tissue of 11, yielding a median TBR of 4.0. Sixteen of the seventeen lesions were adequately sampled in the final pathology. The final pathology was positive for the suspected malignancy in 12 of the 17 excised lesions, 4 of the 17 lesions were benign, and 1 lesion was inadequately sampled. When comparing the malignant and benign lesions, the average ex vivo TBR values were 4.7 and 2.1 with a *p*-value of 0.0181. There was a significant difference between the two groups. This is promising that even without a tumor-specific marker, FDG avidity is more apparent in malignant tissue and is detected by the PET probe.

### 4.1. Safety and Feasibility

In this small subset of patients with pediatric cancer, the PET probe RGS was safe and feasible. The standard PET scan protocol was pre-operatively used for the infusion of IV FDG. The protocol was carried out as planned with no adverse reactions or events identified in the patients who participated in the study. A PET probe intra-operative survey may add a small amount of operative time and cost but were not quantified in the present study. Radiation exposure to patients and perioperative staff from the radiotracer decay is a risk of the procedure. The literature has demonstrated that FDG PET/CT has been associated with a very small radiation exposure risk to perioperative personnel, about 4 µSv of exposure per FDG administration, which is equivalent to that of less than a single-plain X-ray film on the chest [28,29,30]. It is associated with a larger risk of radiation exposure to the patient of 6–7 mSv, or roughly the equivalent of two whole-body CT scans [28,29,30]. The radiation exposure from FDG-guided procedures is higher than those utilizing technetium given the high-energy gamma rays emitted by FDG decay [31].

### 4.2. Clinical Implications and Future Research

While the PET probe is inadequate in its ability to guide surgery, it was able to confirm significantly elevated extracorporeal cps in excised lesions and served as a confirmation of FDG-avid tumor excision. More specific agents or detection systems may need to be developed to provide highly specific intra-operative guidance. FDG is not tumor-specific, and simple handheld gamma detection has a limited specificity for the presence of FDG in malignant tissue versus background tissue. The radiolabeling of a tumor-specific agent such as an antibody, known as radioimmunoguided surgery (RIGS), may allow handheld gamma probes to provide intraoperative guidance for tumor resection in pediatric cancer surgery. In 2007, Sun et al. described the use of RIGS for the excision of occult lesions in metastatic colorectal cancer by utilizing a handheld probe in combination with a radiolabeled monoclonal antibody to a glycoprotein complex overexpressed in epithelial-derived cancers [32]. Targeted therapies are gaining enthusiasm in pediatric cancer treatment, and a similar approach that uses some of these agents may allow for intraoperative guidance for the resection of pediatric tumors.

### 4.3. Strengths and Limitations

Our study is limited in its lack of broad applicability as the operations analyzed were conducted by a single pediatric surgeon experience in pediatric oncology surgery at a large, academic children’s hospital. We did not evaluate the effect of prior radiation and the number of prior operations on the success or duration of the excisions. PET probe performance can be impacted by tumor histology, FDG avidity, and size, as well as the timing of operative exploration after FDG injection and anatomic location. The PET probe demonstrated a consistent performance over broadly dispersed tumor histologies and variable anatomic locations, including cervical, intra-abdominal, and intra-thoracic operations. We did not quantify the effect of these factors on outcomes. This could be examined in future studies. Due to the low number of patients in our series, our data are insufficient for drawing comprehensive conclusions about the specificity of handheld PET probes in detecting malignant tissue. Additionally, post-operative PET scans were not reviewed to confirm the complete excision of residual disease. As with other investigations of rare diseases, it will likely be necessary for cooperative trials to be conducted with the amalgamation of multicenter data on pediatric FDG-guided surgery, in order to draw conclusions about the efficacy of the PET probe. Ideal future studies involving RIGS could potentially improve probe specificity for malignant tissue and allow for intra-operative PET probe localization.

## 5. Conclusions

We demonstrated a successful excision of occult FDG-avid tumors in eight pediatric patients utilizing PET probe RGS. To our knowledge, this is the first case series to describe FDG-guided surgery in a pediatric patient population and the first study to evaluate the PET probe using a PET scan calibration phantom. Tumor localization is limited by lack of FDG specificity and uptake by adjacent metabolically active organs in the thoracic and abdominal cavities. The median TBR of 4.0 is promising and demonstrates that, with improved tissue specificity by novel radiolocalizing agents, a PET probe could potentially be used for in vivo localization in the future. The handheld PET probe has the potential to facilitate a successful tumor excision for diagnostic and therapeutic purposes in pediatric cancer surgery.

## Figures and Tables

**Figure 1 cancers-14-02221-f001:**
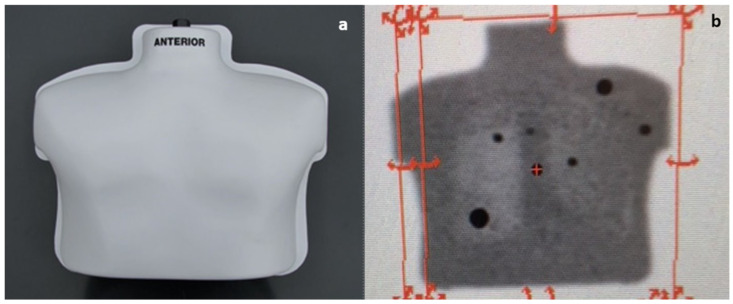
PET calibration phantom model. (**a**) Thoracic phantom model; (**b**) appearance of calibration model on PET-CT demonstrating 6 FDG containing spheres in a dilute FDG solution.

**Figure 2 cancers-14-02221-f002:**
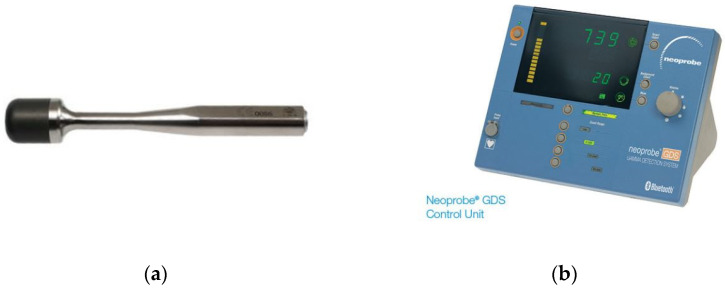
PET Probe System for clinical intra-operative use. (**a**) Hand-held probe capable of detecting high-energy photons, (**b**) Neoprobe gamma detections system control unit with gamma counts per second (cps) readout (Mammotome, Cincinnati, OH, USA).

**Figure 3 cancers-14-02221-f003:**
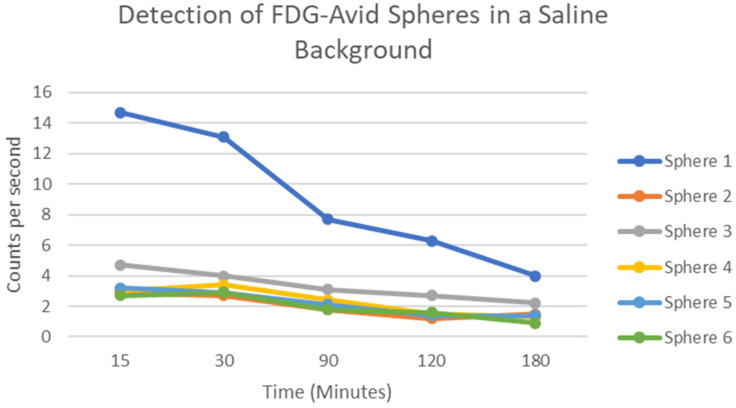
PET calibration phantom model demonstrating FDG decay over time (half-life 110 min).

**Table 1 cancers-14-02221-t001:** Tumor to background ratio of simulated tumors in dilute FDG phantom model. FDG-containing spheres were loaded into a fluid-filled model containing fluid with dilute FDG to a 4:1 SUV at 110 min. This demonstrated that TBR decreased when lesions were within a metabolically active background. Median TBR 1.07.

Measurement	Background	Lesion 1	Lesion 2	Lesion 3	Lesion 4	Lesion 5	Lesion 6
cps	224.1	239.7	246.8	238.1	249.1	238.7	229
TBR	NA	1.07	1.1	1.06	1.11	1.07	1.02

TBR (Tumor-to-background ratio); cps (gamma counts per second as measured by PET probe).

**Table 2 cancers-14-02221-t002:** Patient demographics.

Demographic Characteristics	Total (*n* = 8)
Age (years), median (range)	16 (5–21)
Male gender, *n* (%)	6 (75%)
Patient weight (kg), median (range)	57 (16–96)
Minimally invasive operation, *n* (%)	5 (55%)

**Table 3 cancers-14-02221-t003:** PET probe intra-operative external survey. External PET probe survey of metabolically active solid organs, distal extremity, and room background.

Operation	Liver (cps)	Spleen (cps)	Bladder (cps)	Kidney (cps)	Brain (cps)	Distal Extremity (cps)	Room Background (cps)
1	150	170	-	-	-	-	-
2	392	386.6	2416	370	1358.8	43.7	1.6
3	518	418	897	-	918	48	2
4	312	475	237	-	662	50	-
6	506	467	1267	355	816	96	0.8
7	284	246.3	929	-	1144	53.1	1.4
8	520	426	1941	402	882	64	-
Median	392	418	1098	373	882	50	2

**Table 4 cancers-14-02221-t004:** Prospective PET probe data from nine pediatric cancer operations.

Excised Lesion	Suspected Diagnosis	Location of Lesion	Operative Approach	Lesion Ex Vivo (cps)	Background Tissue Ex Vivo (cps)	TBR	Final Pathology
1	PTLD	Abdomen	MIS	40	11	3.6	PTLD
2	-	Abdomen	-	20	11	1.8	PTLD
3	PTLD	Abdomen, Pelvic Side Wall	MIS	212	137	1.5	PTLD
4	-	Abdomen, Colon	-	309	137	2.3	PTLD
5	Hodgkin’s Lymphoma	Cervical, Supraclavicular	Open	223	19	12.0	Hodgkin’s Lymphoma
6	Hodgkin’s Lymphoma	Mediastinum, Cardiophrenic	MIS	33	9	3.7	Hodgkin’s Lymphoma
7	-	Mediastinum, Anterior	-	8	9	0.9	Inadequate
8	Neuroblastoma	Retroperitoneal Mass	Open	18	5	3.6	Negative
9	-	Retroperitoneal Mass	-	32	5	6.4	Neuroblastoma
10	-	Retroperitoneal Mass	-	20	5	4.0	Neuroblastoma
11	-	Retroperitoneal Mass	-	18	5	3.6	Neuroblastoma
12	-	Retroperitoneal Mass	-	17	5	3.4	Neuroblastoma
13	PTLD	Thoracic, Lung	MIS	32	5	6.9	PTLD
14	Burkitt’s Lymphoma	Abdomen, Small Bowel	MIS	297	223	1.3	Negative
15	PTLD	Cervical, Posterior	Open	15	12	1.3	Negative
16	-	Cervical, Posterior	-	33	12	2.7	Negative
17	Hodgkin’s Lymphoma	Cervical, Lateral	Open	256	36	7.1	Hodgkin’s Lymphoma
Median	-	-	-	32	11	4.0	-

## Data Availability

The data presented in this study are available on request from the corresponding author. The data are not publicly available due to data involving Protected Health Information.

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
