# Peer review of "Handheld PET Probe for Pediatric Cancer Surgery"

_cancers, 2022, doi:10.3390/cancers14092221_

Round 1
Reviewer 1 Report
The present manuscript provides a characterization of the radio-guided surgery with the PET probe, which is of interest. There seems to, however, be a lack of detail in some important aspects.
(1)
Page 2
The authors mention the PET calibration phantom design for the evaluation of the PET probe. However, there is no detailed explanation. To understand the performance of the PET probe, the author should explain “varying size” of six spheres and the concentration of FDG solution in Section 2.1. In addition, please describe the detailed information, such as the size, product name, and so on, about PET calibration phantoms.
(2)
Figure 2 (page 4)
The purpose of the figure is not clear and there are some aspects that the reviewer cannot understand. The intrinsic decay of FDG has been already revealed previously, meaning that the result shown in Fig. 2 may not be new, because the half life of 110 min is clear as mentioned by the author. Could you clearly explain the purpose of Fig. 2?
The author should add the error bar and the measurement time point in Fig. 2. The reviewer concerns the error value of the measurement because the small value is measured at 30 min for Sphere 4(?). It’s hard to see the color difference of the 6 lines showing the results of spheres 1-6 in Fig. 2. Please use different line profiles, such as solid, dotted, and so on.
Why the sphere 1 shows the largest count among 6 samples?
(3) Page 6, line 211
The author claim that the external survey is an interesting finding. However, there is no description regarding the reason why the finding is interesting. In addition, the reviewer cannot understand the reason why the brain shows the second-largest count and cannot find the information about the injection site. Could you explain that?
(4)
Section 4.3 and Introduction
The reviewer concerns that a radioactive method involves a clear limitation due to the handling and facilities of the radioisotope. Recently, T. Kurita et al. demonstrated a novel method without radioactive materials, “magnetically guided localization using a guiding-marker system and a handheld magnetic probe” (https:// doi.org/10.3390/cancers13122923). To enhance the significance of your paper, the reviewer strongly suggests that the author compared your system with the magnetic system, for strength and limitations.
Minor comments;
Page 2, line 90
What is SUV? Please use the full name when the word appears for the first time.
Page 7, line 257
The resolution is “spatial resolution”? The word “resolution” is not clear.
Page 7, line 266
“To this author’s knowledge” should be replaced with “To our knowledge”.
Reviewer 2 Report
The manuscript by the authors provide results from experiments from a PET probe used for cancer surgery. However, there are a few important points needing to be addressed before recommending publication.
- In the introduction please provide a brief review on the different types of imaging probes used for cancer. For example below are few worth mentioning.
Fernandes DA, Kolios MC. Zonyl FSP fluorosurfactant stabilized perfluorohexane nanoemulsions as stable contrast agents. 2019 IEEE International Ultrasonics Symposium (IUS): IEEE; 2019. p. 2267-70.
- Please provide some images for the results on the tumor to background ratio values to show how effective the probes are as there are many low values reported.
- More details should be provided in the materials and methods section on how the probes were developed and produced.
Reviewer 3 Report
This is an interesting brief report manuscript aimed to evaluate the performance of a handheld PET probe for the detection of pediatric cancer surgery.
My main suggestion is to run Mann-Whitney tests instead of unpaired Student t tests, and to express data as median +/- range rather than as mean +/- standard deviation values. This seems safer than using parametric statistics because of the small sample size - actually nonsignificant p-values were found in group comparisons using parametric ANOVA, whereas nonparametric statistics might be more powerful and eventually detect some statistical differences.
All the resulting p-values should be reported in the Results section. The Discussion section might require to be modified based on the new data, as well as the Abstract.
Round 2
Reviewer 1 Report
The author's letter and the revised version of the manuscript thoroughly address all my concerns. I consider the revised version of the manuscript as suitable for publication in Cancers.
Reviewer 3 Report
Thank you. No further comments.